# *Staphylococcus aureus*-induced immunosuppression mediated by IL-10 and IL-27 facilitates nasal colonisation

**Alanna M. Kelly**[1], **John M. Leech**[1], **Sarah L. Doyle**[2,3], **Rachel M. McLoughlin**[1]*

**1** Host-Pathogen Interactions Group, School of Biochemistry and Immunology, Trinity Biomedical Sciences Institute, Trinity College Dublin, Dublin, Ireland, **2** Department of Clinical Medicine, School of Medicine, Trinity College Dublin, Dublin, Ireland, **3** Trinity College Institute of Neuroscience, Trinity College Dublin, Dublin, Ireland

* rachel.mcloughlin@tcd.ie

**Data Availability Statement:** All relevant data are within the manuscript and its Supporting Information files.

## Abstract

*Staphylococcus aureus* persistently colonises the anterior nares of a significant proportion of the healthy population, however the local immune response elicited during *S. aureus* nasal colonisation remains ill-defined. Local activation of IL-17/IL-22 producing T cells are critical for controlling bacterial clearance from the nasal cavity. However, recurrent and long-term colonisation is commonplace indicating efficient clearance does not invariably occur. Here we identify a central role for the regulatory cytokine IL-10 in facilitating bacterial persistence during *S. aureus* nasal colonisation in a murine model. IL-10 is produced rapidly within the nasal cavity following *S. aureus* colonisation, primarily by myeloid cells. Colonised IL-10$^{-/-}$ mice demonstrate enhanced IL-17+ and IL-22+ T cell responses and more rapidly clear bacteria from the nasal tissues as compared with wild-type mice. *S. aureus* also induces the regulatory cytokine IL-27 within the nasal tissue, which acts upstream of IL-10 promoting its production. IL-27 blockade reduces IL-10 production within the nasal cavity and improves bacterial clearance. TLR2 signalling was confirmed to be central to controlling the IL-10 response. Our findings conclude that during nasal colonisation *S. aureus* creates an immunosuppressive microenvironment through the local induction of IL-27 and IL-10, to dampen protective T cell responses and facilitate its persistence.

## Author summary

Nasal colonisation by the bacterium *Staphylococcus aureus* is a very common occurrence in the human population. However there is a lack of knowledge on the immune response that controls nasal colonisation. It is known that a local pro-inflammatory immune response is important for bacterial clearance, however sustained colonisation is common-place suggesting efficient clearance may not be occurring. Here we demonstrate for the first time that *S. aureus* is manipulating the host immune response by promoting immuno-suppression in the nasal cavity which enables bacterial survival. We found that the reg-ulatory proteins IL-10 and IL-27 are central to this suppressive response and result in

**Funding:** This research was funded by a Wellcome Investigator Award (202846/Z/16/Z) and a Science Foundation Ireland Investigator Award (15/IA/3041) to RMM (https://wellcome.org/) (https://www.sfi.ie/), and an Irish Research Council government of Ireland postgraduate scholarship (GOIPG/2017/1452) to AMK (https://research.ie/funding/goipg/). The funders had no role in study design, data collection and analysis, decision to publish, or preparation of the manuscript.

**Competing interests:** The authors have declared that no competing interests exist.

reduced protective T cell responses. We also demonstrate that *S. aureus* is inducing IL-27 production to enhance IL-10 production in order to prolong bacterial colonisation. Our findings show that the host-pathogen interaction during nasal colonisation is more complex than previously described and that *S. aureus* is capable of manipulating the regulatory immune response of the host for its' own benefit.

## Introduction

*Staphylococcus aureus* is a major component of the natural human microbiota, colonising the anterior nares of between 15–25% of the human population persistently [1,2]. Persistent colonisation increases the risk for invasive infection particularly when individuals are immune-compromised such as in the hospital setting [3,4]. However, despite this increased risk, only a minority of nasally colonised individuals actually suffer any adverse effects from their co-existence with *S. aureus* indicating that the majority of this organism's time and energy is overwhelmingly directed not at causing invasive disease but rather in maintaining colonisation. To achieve this, it must maintain a particular type of interaction with the human immune system which to-date remains incompletely understood. Unravelling the sophisticated interplay between *S. aureus* and the host immune system that facilitates colonisation is necessary to identify novel decolonisation strategies but also to understand the downstream consequences of colonisation on the host immune response during infection, and potentially vaccination.

Polymorphisms in innate immune genes have been identified in persistent carriers [5–7] indicating that a strong innate immune response is critical for effective clearance of *S. aureus* from the nares. Consistent with this, nasal secretions from persistent carriers had weaker antimicrobial activity and were less damaging to *S. aureus in vitro* which may aid in the creation of a more permissive environment for bacterial persistence [8,9], while carrier strains have the capacity to downregulate the expression of human β-defensin-3 by nasal epithelial cells [10], suggesting that *S. aureus* can directly manipulate aspects of the host innate immune system in order to persist in the nasal cavity. Neutrophils are central to the initial immune response elicited during *S. aureus* nasal colonisation and are recruited to the nasal cavity upon *S. aureus* colonisation [11]. Neutrophil depletion prior to and during colonisation resulted in impaired bacterial clearance [11]. However, a robust adaptive response is also crucial for controlling bacterial burden within this tissue. Colonised SCID mice fail to clear *S. aureus* from the nasal cavity even after 28 days by which time there was no longer any bacteria detectable in the nasal cavity of WT mice [11]. While an association between antibodies and *S. aureus* colonisation has not been defined [12,13], a T-cell-mediated response appears to be pivotal in controlling *S. aureus* persistence within the nasal cavity [14,15]. IL-17-deficient mice exhibited significantly increased bacterial burdens within the nasal cavity during *S. aureus* colonisation due to reduced local AMP production and reduced neutrophil accumulation [11,15], while IL-22 has also been shown to play a central role in bacterial clearance during *S. aureus* nasal colonisation through this cytokines ability to promote AMP production and reduce expression of loricrin and cytokeratin 10, host ligands used by *S. aureus* to initiate colonisation [14]. Observations in humans also suggest that a robust T cell response is required to prevent *S. aureus* nasal colonisation. Children and adults with HIV who have lower numbers of circulating CD4+ T cells have an increased prevalence of *S. aureus* colonisation [16,17]. Taken together evidence to-date suggests that the host mounts an inflammatory response to attempt to eradicate *S. aureus* from the nasal cavity during colonisation, with the T cell cytokines lL-17 and IL-22 central to

this process, however *S. aureus* may be capable of counteracting these responses to promote persistence.

The subversion of host anti-inflammatory immune responses is an important virulence strategy employed by many pathogens to prevent clearance from the host [18]. In particular, a number of pathogens that infect the respiratory tract, such as *Mycobacterium tuberculosis* [19], and *Klebsiella pneumoniae* [20] have the capacity to exploit host-derived IL-10 for their own benefit to facilitate persistence. Interestingly, *Streptococcus pneumoniae* promotes the induction of TGF-β and the expansion of T-regs within the nasopharyngeal tissue to facilitate persistent colonisation [21], supporting the notion that respiratory tract commensal species, like their intestinal counterparts [22,23], exert immunosuppressive responses to facilitate survival within their niche. It has previously been shown that *S. aureus* promotes IL-10 responses in order to persist during chronic systemic infection and biofilm models of infection [24,25]. While during acute systemic infections, IL-10 appears to aid in host protection by preventing hyperinflammatory responses. During acute systemic infection IL-10 is rapidly expressed by CD19+CD11b+CD5+ B1a regulatory cells in a TLR2-dependent manner and acts to controls excessive inflammation by dampening down effector T cell responses to prevent systemic immunopathology [26]. In contrast, during localized acute skin infection IL-10 acts to promote prolonged bacterial survival through the same mechanisms of impeding protective T cell-associated responses [26]. It appears therefore that at certain anatomical sites *S. aureus* can manipulate IL-10 levels into excess which suppresses otherwise protective T cell responses, thus facilitating persistence of the bacterium to the detriment to the host. Whether *S. aureus* can induce immune tolerance as a mechanism by which to facilitate its survival during colonisation remains to be established.

In this study we define for the first time a central role for IL-10 in the local host immune response elicited during *S. aureus* nasal colonisation. We demonstrate that IL-10 is rapidly produced by myeloid cell populations along with B cells in the nasal cavity upon *S. aureus* colonisation. This IL-10 acts to dampen protective local inflammatory T cell responses which in turn leads to inefficient clearance of *S. aureus* from the nasal cavity. IL-10$^{-/-}$ mice demonstrate enhanced clearance of *S. aureus* from their noses and this is associated with higher numbers of IL-17+ and IL-22+ T cell populations within the tissue in comparison to WT mice. Furthermore, we demonstrate that IL-10 is not the sole mechanism of immunosuppression induced by *S. aureus* to facilitate persistence during colonisation. The regulatory cytokine IL-27 is also induced locally within the nasal tissue and acts upstream of IL-10, promoting its' production. Our findings identify the importance of *S. aureus* mediated immunosuppression in enabling *S. aureus* persistence in the commensal setting.

## Results

### *S. aureus* drives IL-10 expression from innate immune cells within the nasal cavity during colonisation

To establish if IL-10 is induced locally within the nasal cavity during colonisation with *S. aureus*, WT C57BL/6 mice were colonised with streptomycin-resistant *S. aureus* strain Newman (Newman Sm$^R$) at 2x10$^8$ CFU/nose. Over the course of 7 days IL-10 mRNA and protein levels were assessed in both the nose and nasopharyngeal tissue in comparison to control mice administered PBS only. *S. aureus* has been previously shown to colonise both the anterior nares (nose) [14,15] and the deeper nasopharyngeal tissues (NT) [27]. IL-10 gene expression was significantly upregulated in the NT by 6h post-colonisation (Fig 1A) with a similar trend seen in the nose (Fig 1B). IL-10 expression at the protein level was also significantly increased

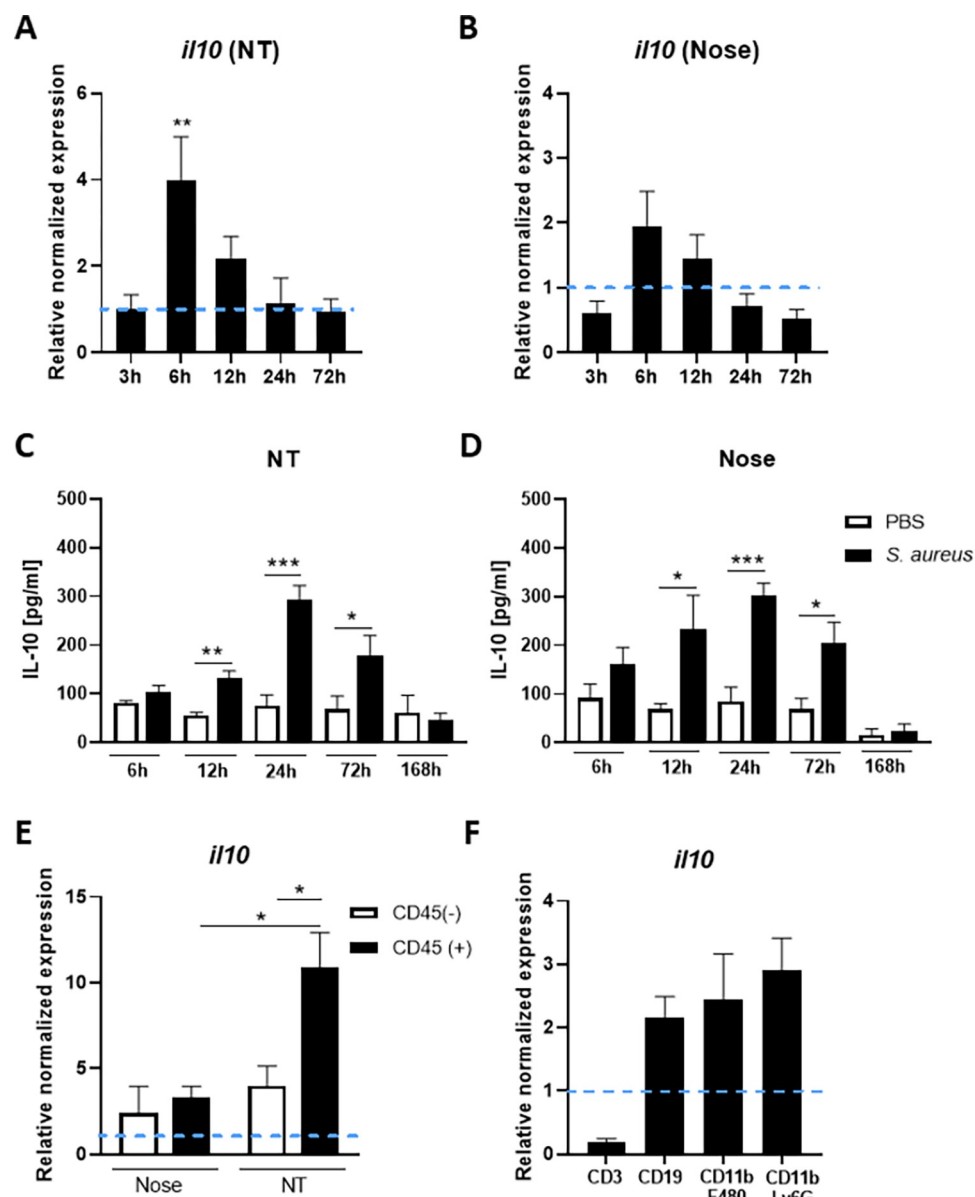

**Fig 1. Interleukin (IL)-10 is upregulated in the nasal cavity during *S. aureus* colonisation.** Wild-type mice were colonised with *S. aureus* Newman $Sm^R$ ($2 \times 10^8$ colony-forming units/nose) or administered with PBS only. At 3h, 6h, 12h, 24h, and 72h mice were culled, tissue was homogenized and RNA extracted from nasopharyngeal tissue (NT) and nose homogenates. IL-10 gene expression in the NT (A) and nose (B) were established using qRT-PCR. The messenger RNA values were expressed as mean relative expression ± s.e.m. and was compared with baseline IL-10 expression from PBS-treated mice after normalizing to 18S RNA expression (Experimental unit = 1 mouse, *n* = 5 for each time-point, total # of animals used; 50, data generated from 2 independent experiments). At 6h, 12h, 24h, 72h and 168h noses and NT were homogenized in PBS and protein levels of IL-10 (C & D) were determined by ELISA. Values are expressed as mean protein concentration ± s.e.m. (Experimental unit = 1 mouse, *n* = 5–10 per group, total # animals used 80, data generated from 2 independent experiments). At 6h post-colonisation noses and NT were excised, tissue digested and CD45+ and CD45- cells isolated by MACs sorting prior to assessment of IL-10 mRNA expression (E) (Experimental unit = Tissue was pooled from n = 10 mice and the experiment was repeated twice, total # animals used 20). The CD45+ fraction was also sorted into CD3+ T cells, CD19+B cells, CD11b+ F480+Ly6G- macrophages and CD11b+ Ly6G+ F480- neutrophils/MDSCs using flow cytometry cell sorting (F). IL-10 mRNA expression was determined by RT-PCR for each fraction (Experimental unit = Tissue was pooled from n = 15 mice and the experiment was repeated twice, total # animals used 30). Statistical analysis was performed using two-way analysis of variance or student t test. *$P \leq 0.05$; **$P \leq 0.01$; ***$P \leq 0.001$.

in both the NT and the nose with IL-10 production peaked at 24h post-colonisation at both sites (Fig 1C and 1D).

To establish the cellular source of this IL-10, noses and NT were excised at 6h post-colonisation and tissue digested prior to isolating CD45+ and CD45- cells. IL-10 gene expression was assessed in the CD45+ and CD45- fractions by RT-PCR. Within the nose IL-10 gene expression was detectable from both immune cells and non-immune cells albeit at very low and relatively equal levels. In contrast, in the NT IL-10 gene expression was significantly increased compared to the nose and was primarily detected in the CD45+ fraction (Fig 1E). CD45+ cells isolated from the NT of colonised mice were then further FACS sorted into CD3+T cells, CD19+B cells, CD11b+F480+Ly6G- and CD11b+F480-Ly6G+ myeloid cells and their RNA extracted. The CD3+T cell population expressed little to no IL-10 mRNA whereas CD19+B cell, CD11b+F480+Ly6G- and CD11b+F480-Ly6G+ cells expressed considerable levels of IL-10 (Fig 1F). It is likely that the CD19+B cells are innate-like B1a regulatory cells which have previously been shown to produce IL-10 in response to *S. aureus* exposure [26]. Consistent with this we were able to detect a small population of CD19+ CD5+B cell, within the nasal cavity of *S. aureus* colonised mice (S1 Fig). Overall this data suggests that IL-10 is produced rapidly in the nasal cavity in response to *S. aureus* nasal colonisation primarily by a variety of innate immune cells likely comprising of innate-like B cells, macrophages, neutrophils and myeloid derived suppressor cells (MDSCs). Consistent with this we demonstrated that bone marrow derived macrophages (BMDMs) were capable of robust IL-10 production in response to in vitro exposure to *S. aureus*. BMDMs were exposed to *S. aureus* strain Newman (S2A Fig) or a panel of *S. aureus* strains that had been isolated from persistent nasal carriers (S3A Fig) for 24 hours and IL-10 production measured by ELISA. All isolates drove comparable levels of IL-10 suggesting that the induction of IL-10 by *S. aureus* during nasal colonisation is not strain dependent. Importantly human colonising isolates were capable of colonising mice at similar levels to *S. aureus* strain Newman (S3B–S3C Fig).

## IL-10 production within in the nasal cavity facilitates persistence of *S. aureus* during colonisation

We then employed IL-10 deficient (IL-10$^{-/-}$) mice to establish the effect that IL-10 was having on persistence of *S. aureus* within the nasal tissue. WT and IL-10$^{-/-}$ mice were intranasally colonised with Newman Sm$^R$ ($2 \times 10^8$ CFU per nose) and bacterial burdens enumerated on day 3, 7 and 10 post-colonisation. Both WT and IL-10$^{-/-}$ mice were initially colonised with *S. aureus* to a similar extend however CFUs were significantly reduced in the IL-10$^{-/-}$ noses compared to WT noses by day 3, post colonisation (Fig 2A) with IL-10$^{-/-}$ mice effectively clearing the bacteria from their noses by day 7 (Fig 2B), whereas in WT animals, bacteria persisted up to 10 days and has previously been shown to persist beyond 14 days in this model [14]. Assessment of bacterial burden was focused on the nose as it has been shown that colonisation at this site dominates in this model [14]. Consistent with this we confirmed significantly higher levels of bacteria colonising the nose, vs. the NT or the lungs (S4 Fig).

## *S. aureus*-induced IL-10 leads to reduced numbers of IL-22+ and IL-17+ T cells in the NT during *S. aureus* nasal colonisation

Previously published data has identified the cytokines IL-17 and IL-22 as particularly important aspects of the host immune response that control *S. aureus* nasal colonisation [11,14,15]. To establish the effect *S. aureus*-induced IL-10 was having on the local production of these cytokines, WT and IL-10$^{-/-}$ mice were intranasally colonised with *S. aureus* and the levels of IL-17 and IL-22 expressing T cell populations within the NT assessed by flow cytometry on

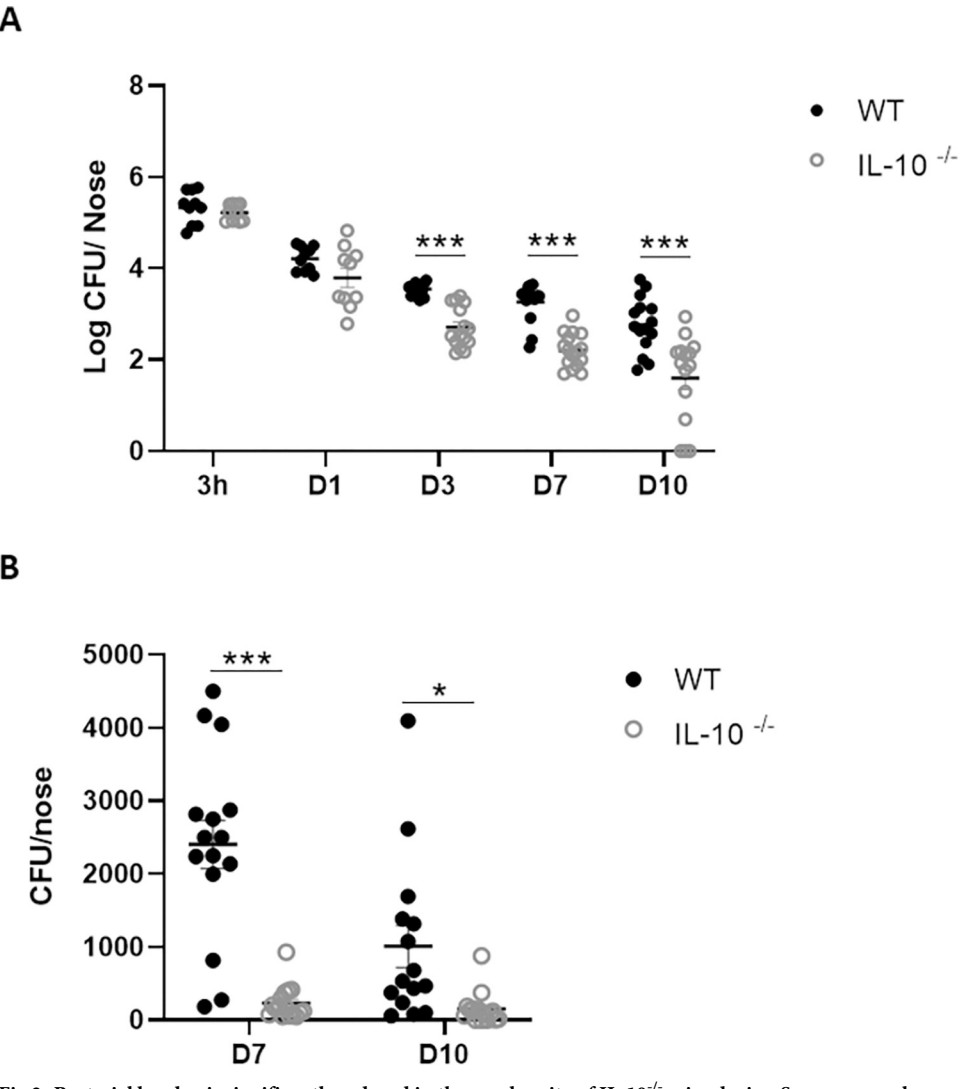

**Fig 2. Bacterial burden is significantly reduced in the nasal cavity of IL-10<sup>-/-</sup> mice during *S. aureus* nasal colonisation.** Wild-type and IL-10<sup>-/-</sup> mice were intranasally colonised with *S. aureus* Newman Sm<sup>R</sup> ($2 \times 10^8$ colony-forming units/nose). At 3h, 1 day, 3 days, 7 days and 10 days mice were culled and noses were excised. Noses were homogenized and serial dilutions of homogenates were plated onto streptomycin-supplemented TSA plates. Plates were grown overnight and CFUs were enumerated. Results are expressed as Log CFU/nose (A) (Experimental unit = 1 mouse, n = 10–16 animals per time point, total # animals used 133, data pooled from 3 independent experiments). Day 7 and Day 10 data are also depicted as CFU/nose (B). Statistical analysis was carried out by one-way ANOVA, and student t-test. *P$\leq$0.05; *** P$\leq$0.001.

day 3 and 7 post-colonisation. IL-17+T cells were increased in the NT in both WT and IL-10<sup>-/-</sup> mice on day 3 and 7 post-*S. aureus* colonisation (Fig 3A,3B and 3C) as compared to PBS-treated mice. Both cytokines have previously been shown to be expressed within this tissue [11,14]. Interestingly, the major source of IL-17 in the NT was shown to be γδ+T cells (Fig 3C), with CD4+T cells (Fig 3A) and a smaller number of CD8+T cells (Fig 3B) also contributing to the IL-17 production. IL-17+T cells peaked on 3 days post-colonisation, and the IL-10<sup>-/-</sup> mice had significantly higher numbers of IL-17 producing CD4+T cells (Fig 3A), CD8+T cells (Fig 3B) and γδ+T cell (Fig 3C) on both day 3 and day 7 compared to WT colonised-mice.

IL-22+T cell subsets were also significantly increased upon exposure to *S. aureus* in both WT and IL-10<sup>-/-</sup> mice compared to the PBS-treated mice (Fig 3D, 3E and 3F). In contrast to

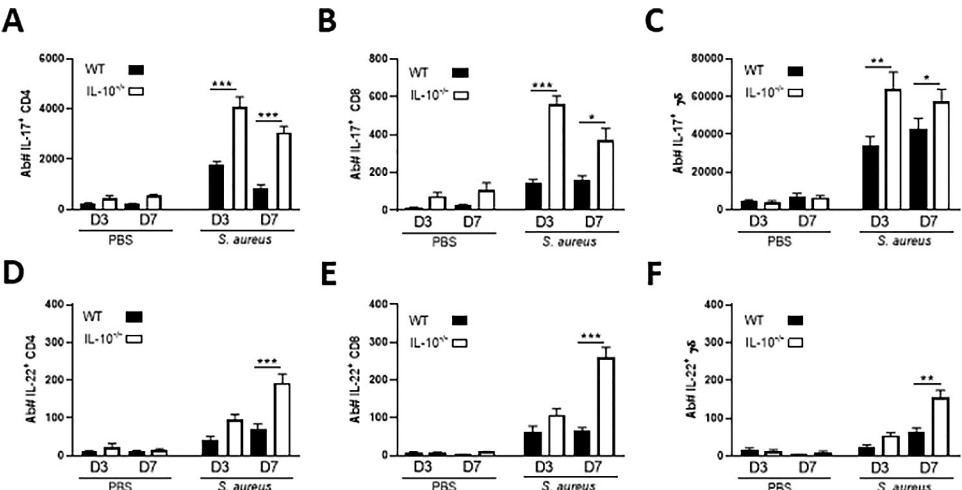

**Fig 3. IL-10 suppresses effector T cell responses in the NT during *S. aureus* colonisation.** WT and IL-10$^{-/-}$ mice were intranasally inoculated with *S. aureus* Newman Sm$^R$ ($2 \times 10^8$ colony-forming units/nose) or administered with PBS only. On 3 days and 7 days mice were culled, the NT was removed, and tissue digested for flow cytometry analysis. Cells were gated on single, live CD45+ cells> CD3+> CD4+/CD8+ or CD4-CD8-, double negatives were gated into γδ +. All T cell subsets were next gated on cytokine production; IL-17+ CD4+ (A), IL-17+CD8 +(B) IL-17+γδ + (C), IL-22 + CD4 + (D), IL-22+CD8 + (E) and IL-22+γδ +(F). Results are expressed in absolute cell numbers (Ab #) with mean ± S.E.M. (Experimental unit = 1 mouse, n = 10–20, per group, total # animals used 130, data generated from 4 independent experiments). Statistical analysis was carried out by two-way ANOVA. ** P≤0.01, *** P≤0.001.

the IL-17+T cells, IL-22+T cells peaked on day 7, indicating that IL-22 is acting at a later time-point than IL-17, which agrees with previously published data [11,14]. The overall numbers of IL-22+T cells present in the NT were lower than IL-17+T cells, with each of the T cells subsets contributing relatively equally to the IL-22 production on day 7 post-colonisation. All IL-22+T cell subsets were significantly elevated in the IL-10$^{-/-}$ mice compared to WT mice colonised with *S. aureus*.

Overall, these results demonstrate that *S. aureus*-induced IL-10 acts to suppress local IL-17 and IL-22 T cell responses within the NT during colonisation. These cytokines then control other aspects of the immune response required to control bacterial persistence within the tissue such as local expression of antimicrobial peptides and neutrophil migration. Consistent with this we demonstrated that neutrophil accumulation within the NT is significantly increased in IL-10$^{-/-}$ as compared to WT mice during *S. aureus* nasal colonisation (S5 Fig).

### *S. aureus* also drives IL-27 expression from innate immune cells within the nasal cavity during colonisation

Having identified that local induction of IL-10 is central to *S. aureus*-induced immunosuppression during nasal colonisation we wanted to establish if IL-10 was acting alone or in combination with other regulatory cytokines. Previous studies have demonstrated that the IL-12 family cytokine IL-27 plays an important regulatory role during pulmonary infection, by suppressing inflammatory immune responses and promoting IL-10 production [28,29]. We therefore investigated whether IL-27 was induced by *S. aureus* within the upper respiratory tract during nasal colonisation. Expression of the IL-27 *p28* subunit and IL-27 protein levels were assessed in both the nose and NT of colonised mice in comparison to non-colonised PBS controls. IL-27 gene expression was already upregulated in the NT at 3h post-colonisation and significantly so at 6h post-colonisation (Fig 4A). By 12h, expression was reduced almost to control levels. Consistent with this IL-27 protein levels were also significantly increased in the

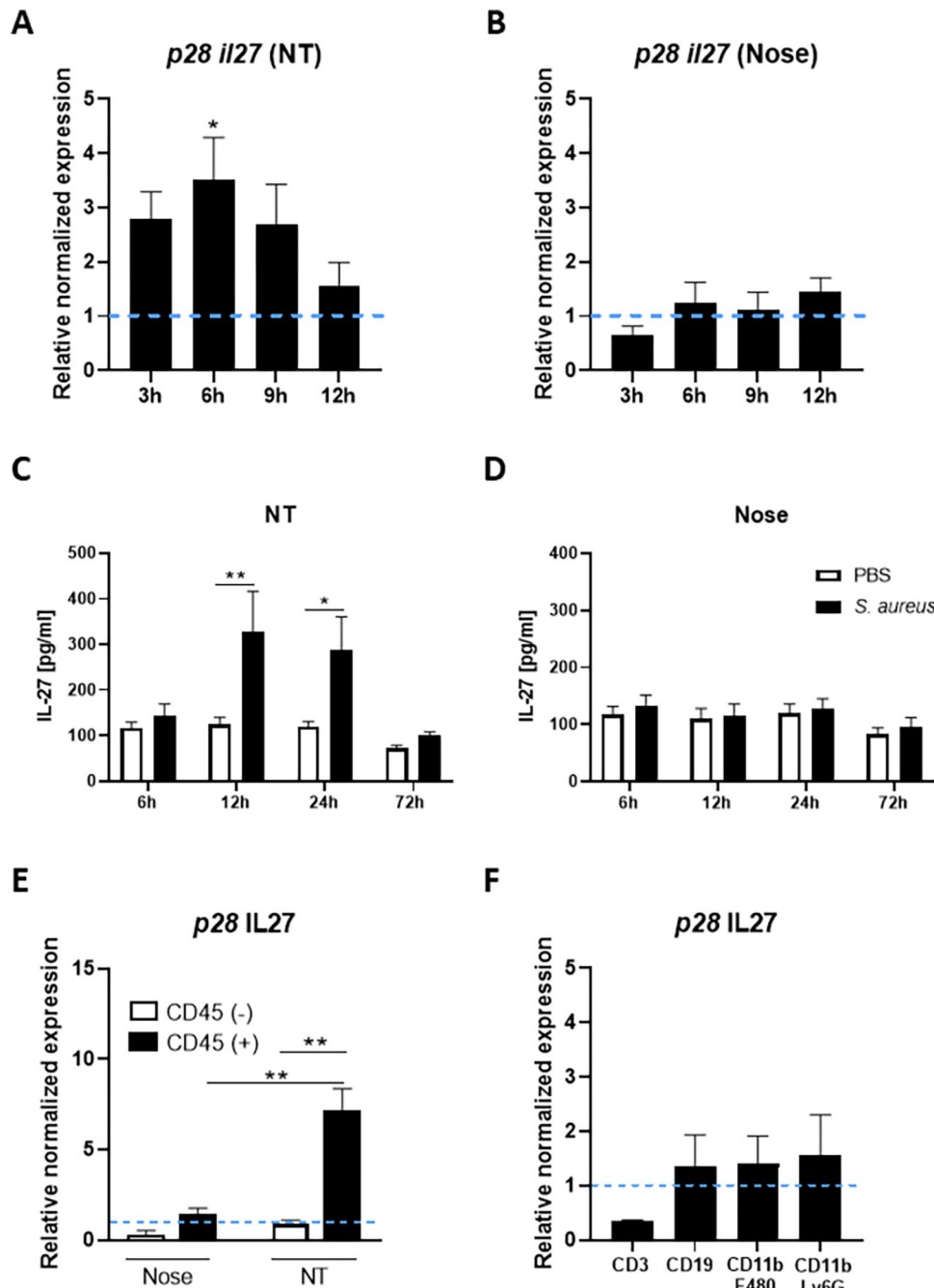

**Fig 4. IL-27 is rapidly upregulated in the nasal cavity during *S. aureus* colonisation.** WT mice were colonised with *S. aureus* Newman Sm$^R$ ($2 \times 10^8$ colony-forming units/nose) or administered with PBS only. At 3h, 6h, 9h and 12h post-colonisation mice were culled, tissue homogenized and RNA extracted from nose and nasopharyngeal tissue (NT) homogenates. IL-27 gene expression in the NT (A) and nose (B) was established using qRT-PCR. The messenger RNA values were expressed as mean relative expression ±s.e.m. and was compared with baseline IL-27 expression from PBS-treated mice after normalizing to 18S RNA expression (Experimental unit = 1 mouse, n = 5, per group, total # animals used 40, data generated from 2 independent experiments). At 6h, 12h, 24h and 72h noses and NT were homogenized in PBS and protein levels of IL-27 (C & D) were determined by ELISA. Values are expressed as mean protein concentration ±s.e.m. (Experimental unit = 1 mouse, *n* = 5–10, per group, total # animals used 70, data generated from 2 independent experiments). At 6h post-colonisation noses and NT were excised, tissue digested and CD45+ and CD45- cells isolated by MACs sorting prior to assessment of IL-27 mRNA expression (E) (Experimental unit = Tissue was pooled from n = 10 mice and the experiment was repeated twice, total # of animals used 20). The CD45+ fraction was also sorted into CD3+, CD19+, CD11b+ F480+ and CD11b+ Ly6G+ F480- using flow cytometry cell sorting (F).

IL-27 mRNA expression was determined by RT-PCR for each fraction (Experimental unit = tissue was pooled from n = 15 mice and the experiment was repeated twice, total number of animals used 30). The same mice were used as in Fig 1. Statistical analysis was performed using two-way analysis of variance or student t test. *$P \leq 0.05$; **$P \leq 0.01$,*** $P \leq 0.00$.

NT at 12h and 24h post-colonisation (Fig 4C) with IL-27 production peaking at 12h post-colonisation, indicating that IL-27 is expressed earlier than IL-10 within this tissue. There was no increase in IL-27 gene or protein expression in the nose of colonised mice (Fig 4B and 4D). To determine the cellular source of the IL-27, the NT was excised at 6h post-colonisation and tissue digested before isolation of CD45+ and CD45- cells by MACs sorting. IL-27 gene expression was then assessed in both fractions and was found to be significantly elevated in cells isolated from the NT as compared to the nose (Fig 4E). Within the NT IL-27 is exclusively being produced by CD45+ cells (Fig 4E). CD45+ cells were then FACS sorted into CD3+T cells, CD19+B cells, CD11b+F480+Ly6G- and CD11b+F480-Ly6G+ myeloid cells (Fig 4F). While the overall level of IL-27 gene expression detected was low, CD19+B cell, CD11b+F480 + and CD11b+Ly6G+ cell populations were all shown to be contributors, with each contributing relatively equal amounts whereas CD3+T cells expressed little to no IL-27. It is possible that there may be alternative CD45+ cells contributing to IL-27 production within the nasal tissue. Previous studies have shown that dendritic cells are important innate cells also capable of producing IL-27 to suppress inflammatory responses [30–32], suggesting that these cells could also contribute to the IL-27 response during *S. aureus* nasal colonisation.

## S. *aureus*-induced IL-27 promotes bacterial persistence within the nasal cavity through the induction of IL-10

Having confirmed that IL-27 was produced rapidly within the nasal tissue in response to *S. aureus* it was important to establish if this IL-27 was having an impact on local IL-10 levels. IL-27 was blocked by the intranasal administration of αIL-27 (150μg/mouse) 24h prior to and during colonisation with *S. aureus*. Control groups were administered an anti-IL-27 isotype (150μg/mouse) or PBS only. Blockade of IL-27 in WT mice led to a significant reduction in IL-10 levels in the NT at 24 hours post colonisation (Fig 5A), reducing levels almost to that of PBS control mice. Importantly, IL-27 blockade also improved bacterial clearance from noses of WT colonised mice on day 3 (Fig 5B) and day 7 (Fig 5C). In the presence of αIL-27 there was a significant reduction in bacterial burden in the nose at day 3 and day 7 compared to control mice administered with an isotype control (Fig 5B and 5C). Importantly however when IL-27 was blocked in IL-10$^{-/-}$ mice there was no further increase in bacterial clearance, indicating that within the nasal tissue IL-27 acts to promote IL-10 which in turn facilitates bacterial persistence.

## IL-27 directly regulates IL-10 production by macrophages

Our data suggests that IL-27 acts in an autocrine manner to promote IL-10 production within the nasal tissue during colonisation. To further interrogate this interaction, we utilised BMDMs in an *in vitro S. aureus* exposure assay given that CD11b+F480+ macrophages were identified as an important source of both IL-10 (Fig 1F) and IL-27 (Fig 4F) within the nasal tissue during *S. aureus* colonisation, and BMDMs exposed to *S. aureus* were confirmed to be capable of producing both cytokines in a dose-dependent manner (S2A and S2B Fig). When BMDMs were pre-treated with increasing concentrations of αIL-27 30 minutes prior to exposure to *S. aureus*, IL-10 production was significantly reduced (Fig 6A). In contrast when BMDM were infected with *S. aureus* and simultaneously treated with recombinant IL-27, IL-

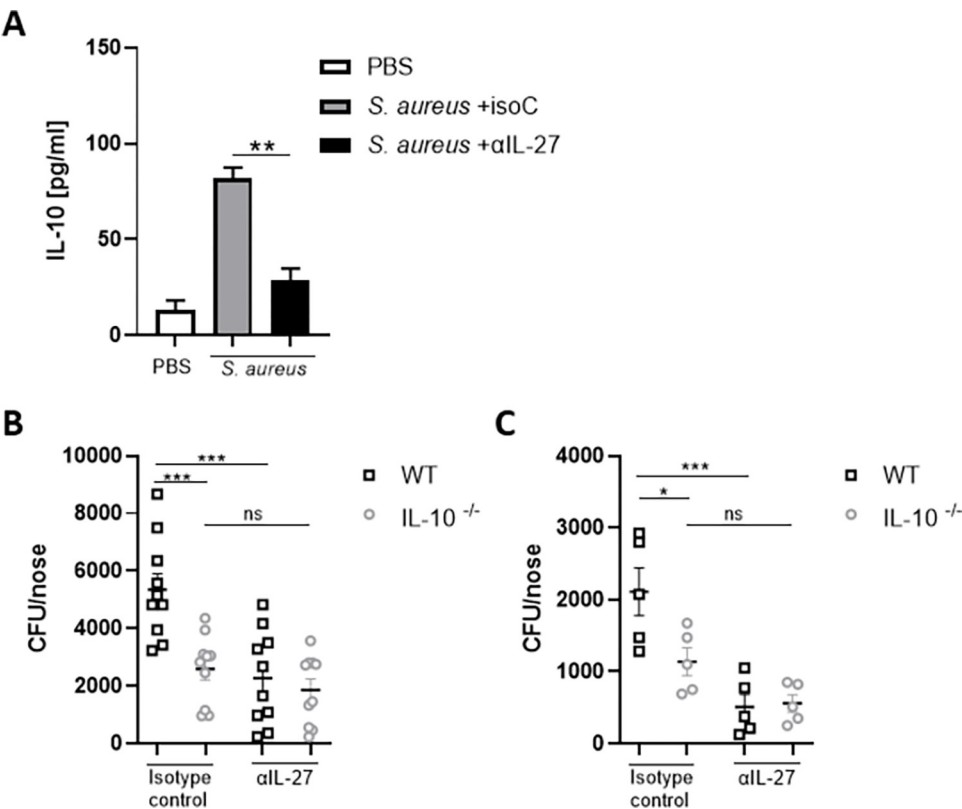

**Fig 5. IL-27 blockade reduces IL-10 production within the nasal tissue and improves bacterial clearance from the nose during *S. aureus* colonisation.** WT and IL-10[-/-] mice were intranasally administered with PBS, αIL-27 (150μg/mouse) or isotype control (150μg/mouse) 24h prior to colonisation. Mice were then colonised with S. *aureus* Newman Sm[R] ($2 \times 10^8$ CFU/nose) in combination with αIL-27 (150μg/mouse) or isotype control (150μg/mouse). At 24h post-colonisation, NT of WT mice was homogenized in PBS and protein levels of IL-10 (A) were determined by ELISA. Values are expressed as mean protein concentration ±s.e.m. (Experimental unit = 1 mouse, *n* = 10 per group, total # animals used 30, data generated from 3 independent experiments). At Day 3 (B) and Day 7 (C) post-colonisation WT and IL-10[-/-] mice were culled and noses excised. Noses were homogenized and serial dilutions of homogenates were plated onto streptomycin-supplemented TSA plates. Plates were grown overnight and CFUs were enumerated. Results are expressed as CFU/nose (Experimental unit = 1 mouse, n = 10 per group, total # animals used 40, data generated from 3 independent experiments) (B) and (Experimenal unit = 1 mouse, n = 5 per group, total # animals used 20, data generated from 2 independent experiments) (C). Statistical analysis was performed using one way analysis of variance or student t test. $^*P \leq 0.05$; $^{**}P \leq 0.01$, $^{***}P \leq 0.001$.

10 production was significantly increased (Fig 6B). This result confirmed that IL-27 can directly promote autocrine IL-10 production by macrophages during *S. aureus* infection. It has previously been shown that *S. aureus* induced IL-10 production by human monocytes is mediated by TLR 2 signalling [33,34]. Here we confirm that IL-10 production in response to *S. aureus* is similarly TLR2-dependent in murine macrophages. WT and TLR2[-/-] BMDMs were exposed to *S. aureus* strain Newman (Fig 6C) or human colonising strains (S6 Fig) and IL-10 production assessed by ELISA. IL-10 production by TLR2[-/-] cells exposed to *S. aureus* was significantly reduced compared to WT cells. Importantly the addition of exogenous rIL-27 did not significantly increase IL-27 levels by the TLR2[-/-] cell (Fig 6C), suggesting TLR2 signalling is the primary signalling pathway controlling IL-10 production in these cells.

## Discussion

Studies have demonstrated that IL-17 and IL-22-mediated CD4+T cell responses are central to the clearance of *S. aureus* during nasal colonisation through these cells ability to regulate

**A**

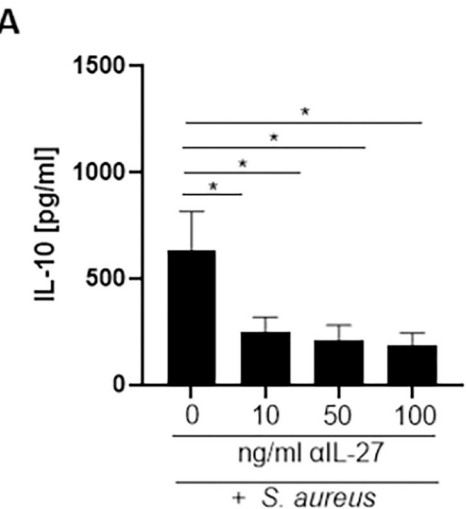

**B**

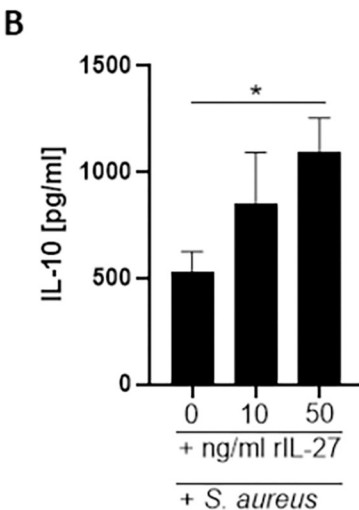

**C**

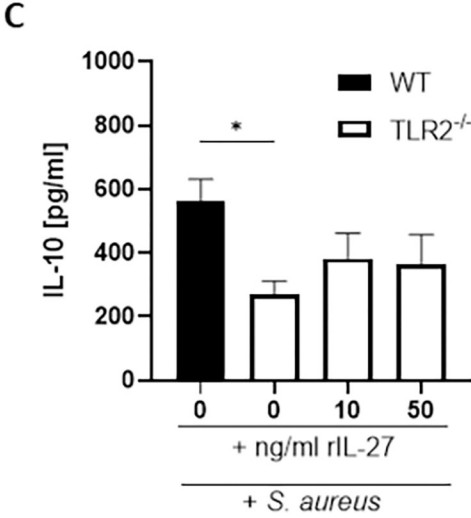

**Fig 6. IL-27 controls IL-10 production in *S. aureus* infected macrophages.** WT bone marrow-derived macrophages (BMDMs) were pre-treated with anti-IL-27 antibody at 10ng/ml, 50ng/ml &100ng/ml for 30minutes prior to exposure to *S. aureus* Newman at MOI 10 for 3 h. Cells were washed with gentamicin to remove extracellular bacteria and were then further incubated for 24 h in the presence of anti-IL-27 antibody (A). WT BMDMs or TLR2-/- BMDMS were pre-treated with recombinant IL-27 at 10ng/ml or 50ng/ml for 30minutes prior to exposure to *S. aureus* Newman at MOI 10 for 3 h. Cells were washed with gentamicin to remove extracellular bacteria and were then further incubated for 24 h in the presence recombinant IL-27 (B, C). The levels of IL-10 were determined by ELISA. Values are expressed as mean protein concentration ± S.E.M. (Experimental unit = BMDMs isolated from 1 mouse, n = 5–6, total # animals used 11, data generated from 6 independent experiments. Statistical analysis was performed using one-way ANOVA. *P≤0.05.

neutrophil recruitment to the nasal cavity, promote local production of antimicrobial peptides and control the expression of staphylococcal-binding ligands within the nose [11,14]. However the fact that many individuals are colonised recurrently and for long periods of time with *S. aureus* [1] suggests that local effector T cell responses are being curtailed in some way so as to prevent efficient elimination of *S. aureus* from the nasal cavity. Our study provides new insights into the mechanisms that underpin *S. aureus* commensal colonisation and indicates that *S. aureus* induced immunosuppression through the local upregulation of IL-10, counteracts local effector T cell responses, resulting in the prolonged survival of the bacterium within the nasal tissue. Myeloid cells and potentially B cells, within the nasal cavity rapidly produce IL-10 upon exposure to *S. aureus*. This IL-10 then acts to dampen down IL-17 and IL-22 producing T cell responses to the detriment of the host. The induction of IL-10 results in reduced bacterial clearance facilitating persistence of *S. aureus* within the nasal cavity. Furthermore, the regulatory cytokine IL-27 is also being induced during *S. aureus* nasal colonisation, again to the advantage of the bacterium. IL-27 acts in an autocrine manner to enhance production of IL-10, further facilitating persistence of the bacterium.

*S. aureus* has been shown to promote IL-10 production to extend its survival during infection [24,26]. Here we describe for the first time that a similar mechanism is being used by the bacterium to survive within the nasal cavity during colonisation. Initial attempts to identify IL-10+ populations within the nasal tissue were carried out using flow cytometric methods however the identification of murine IL-10-producing cells by intracellular staining is challenging given that intracellular IL-10 is typically expressed at a relatively low intensity [35,36]. To circumvent this, we analysed IL-10 expression by individual cell subsets by RT-PCR. B cells and CD11b+ myeloid-derived cells were identified as the major early producers of IL-10 during *S. aureus* nasal colonisation.

Our data is in agreement with previously published data demonstrating CD11b+F480 + macrophages and B cells as important sources of IL-10 during acute systemic and localized infections [26]. The myeloid cell population producing IL-10 within the nasal cavity comprises CD11b+F480+ macrophages but also CD11b+F480+Ly6G+ cells which are most likely MDSCs. Previous studies have demonstrated that immunosuppressive responses mediated by MDSC-derived IL-10 are critical to bacterial persistence during *S. aureus* biofilm infection [24,37]. It has been debated that *S. aureus* may form biofilms in the nasal cavity to facilitate nasal colonisation and persistent survival. The production of binding factors important in the initial phases of biofilm production are upregulated during nasal colonisation [38], while scanning electron microscopy has also confirmed that hallmark biofilm structures are formed by the bacterium within the nasal cavity of mice following colonisation with *S. aureus* strain UAMS-1 [39]. Studies have also demonstrated the potential for immunosuppressive neutrophils to contribute to IL-10 immune responses during pulmonary bacterial infection [40,41] indicating that the IL-10 producing CD11b+Ly6G+ population identified within the nasal cavity has the possibility of also containing regulatory neutrophils. Previous work however

suggests that the IL-10 response to *S. aureus* is predominantly mounted by monocyte/macrophages [33], and that in this cell type sensing of *S. aureus* PAMPs can lead to both pro and anti-inflammatory responses that can be un-coupled in particular in response to colonising isolates [42].

We confirmed using bone marrow-derived macrophages that not only IL-10 but also the regulatory cytokine IL-27 was upregulated when these cells are exposed to *S. aureus*. The induction of the regulatory cytokine IL-27 seems to be a key strategy employed by *S. aureus* to further facilitate persistence during nasal colonisation. IL-27 is rapidly upregulated mainly by myeloid cells within the NT upon exposure to *S. aureus*, where it acts upstream of IL-10 to enhance its' production. A previous study has reported that during *S. aureus* secondary co-infection post influenza infection, IL-27 produced in response to the virus increased susceptibility to *S. aureus* pneumonia in an IL-10 dependent manner [43]. IL-27$^{-/-}$ mice displayed significantly decreased IL-10 production compared to WT mice upon infection with influenza, which led to enhanced bacterial clearance alongside decreased IL-10 production and higher Th-17 associated responses. Our results now indicate that *S. aureus* can itself induce IL-27 production within the nasal tissue and that blocking IL-27 during nasal colonisation results in impeded IL-10 production and significantly improved bacterial clearance. Although IL-10 levels were significantly reduced in the presence of the IL-27 blocking antibody they were not reduced to baseline likely due to direct induction of IL-10 through innate sensing of *S. aureus* PAMPs. Critically, however, clearance of bacteria was not further enhanced when IL-27 was blocked in IL-10$^{-/-}$ mice, demonstrating that during *S. aureus* nasal colonisation the primary function of IL-27 is to enhance IL-10 responses.

IL-27 can act in an autocrine manner to promote IL-10 production in macrophages through the activation of the transcription factors Signal transducer and activator of transcription 1 (STAT 1) and STAT 3 which bind to the IL-10 promoter [44]. Blockade of IL-27 signalling in BMDM during *in vitro S. aureus* exposure led to a significant reduction in IL-10 production while the addition of exogenous IL-27 further enhanced the IL-10 response to *S. aureus* in these cells suggesting IL-27 is also acting in an autocrine manner in this context. The addition of exogenous IL-27 to TLR2$^{-/-}$ BMDMs exposed to *S. aureus* however could not return IL-10 production to WT levels indicating that TLR2 signalling was central to the IL-10 response in these cells. Holley et al. similarly demonstrated reduced IL-10 production by TLR2$^{-/-}$ microglia cells, despite the fact that IL-12 family cytokines, including IL-27, were over-expressed by these cells in response to *S. aureus* [45]. Consistent with this our data suggests that when TLR2 signalling is ablated IL-27 cannot enhance IL-10 production indicating this signalling pathway is ultimately needed for IL-10 production in myeloid cells.

CD4+T cell-derived IL-22 and IL-17 responses have previously been implicated as critically important aspects of the immune response that control *S. aureus* nasal colonisation [11,14]. However, for the first time we demonstrate that γδ+T cells are major producers of these inflammatory cytokines within the nasal cavity and in fact γδ+T cells were identified as the main IL-17 T cell source on both Day 3 and Day 7 post-colonisation. A growing literature highlights the importance of IL-17+γδ+T cells in anti-*S. aureus* immunity [46,47] and furthermore have demonstrated γδ+T cells can also become long-lived memory cells which are capable of permanently populating local anatomical sites post-infection and ensure pathogen control thereafter [48]. Herein we provide the first documented evidence that γδ+T cells may be playing a protective role during *S. aureus* nasal colonisation. A large γδ+T cell population is present in the nasal mucosa of allergic rhinitis patients and are expanded in influenza infection models [49,50], suggesting that these cells can hone to the nasal mucosa to act as a first line defence against infection. In a S*treptococcus pneumoniae* lung infection model IL-17+γδ+T cells were expanded in the NALT of infected mice and were essential to the protective immune

response against subsequent lung infection [51]. Further studies are required to determine if the IL-17+γδ+ T cell populations expanded in response to *S. aureus* nasal colonisation are forming a localized memory population and how their expansion or activity might be manipulated by S. *aureus*-induced IL-10 responses in order to promote bacterial survival.

This study presents a model whereby S. *aureus* targets the anti-inflammatory arm of the local nasal immune system during colonisation to facilitate bacterial persistence. This creation of an immunosuppressive microenvironment dampens inflammatory T cell responses which in turn likely results in impaired neutrophil recruitment and antimicrobial peptide production thus preventing effective bacterial clearance. Previous studies in humans have shown that PBMCs isolated from persistent carriers of *S. aureus* express increased IL-10 levels concomitant with reduced IFN-γ production following exposure to *S. aureus* in vitro [52], while PBMCs isolated from persistent carriers also produced increased levels of IL-19, an IL-10 cytokine family member, upon exposure to endogenous colonising strains vs. non-endogenous strains [53]. Consistent with this we demonstrate that *S. aureus* strains that were isolated from persistent nasal carriers were capable of inducing robust IL-10 response in macrophages further increasing the relevance of this study to the human situation. Other regulatory cytokines such as IL-35 have also been implicated in regulating the local nasal inflammatory response to *S. aureus* within human nasopharynx-associated lymphoid tissue where this cytokine acts to suppress *S. aureus* driven Th17 responses [54]. Our results now provide mechanistic insights into the strategies employed by *S. aureus* to facilitate persistent colonisation and provide proof of concept that by impeding local bacterial-induced immunosuppression by blocking IL-10 and/or IL-27 it may be possible to promote an enhanced local pro-inflammatory immune response and thus improve bacterial clearance. The blockade of IL-27 or IL-10 within the nasal cavity may be a novel beneficial therapeutic treatment for nasal decolonisation. Although any interference with the anti-inflammatory immune response would need to be both fastidious and temporal in order to prevent hyper-inflammation and off-target effects.

Overall, it will be essential to establish the long-term impact that the formation of an immunosuppressive microenvironment by *S. aureus* nasal colonisation is having on host immunity. The creation of this anti-inflammatory setting is likely occurring early in life as nasal colonisation can occur within days of birth [55]. *S. aureus* potentially begins to "program" the local immune cells from this time onwards; therefore, any subsequent exposure such as during an infection may elicit a somewhat ineffective pro-inflammatory immune response and increase the host's risk of serious infection and bacterial spread. This would explain the association between nasal colonisation and the increased risk of acquiring an invasive *S. aureus* infection [2]. However, for the vast majority of individuals asymptomatic carriage has not been proven harmful to the host and there is some evidence to suggest that it could even be protective against severe outcome following invasive disease [56]. Mortality rates from *S. aureus* bacteriemia tend to be lower in persistent carriers, which may be because of their "suppressed" inflammatory responses which do not result in cytokine storm or hyperinflammation. Further studies are therefore warranted to establish the implications of this early immunosuppressive programming of the immune response for subsequent pathogenic exposure during infection and/or responses to vaccination.

## Materials & methods

### Ethics statement

All animal experiments were conducted in accordance with the recommendations and guidelines of the health product regulatory authority (HPRA), the competent authority in Ireland

and in accordance with protocols approved by Trinity College Dublin Animal Research Ethics Committee. Project authorisation number AE19136/P095. Euthanasia by $CO_2$ inhalation.

## Mice

C57BL6J wild-type mice were bred in house at the Trinity College Dublin Comparative Medicine Unit. IL-10$^{-/-}$ mice (C57BL6J background) and TLR2$^{-/-}$ mice (B6.129-*Tlr2$^{tm1Kir}$*/J, C57BL6J background) were obtained from The Jackson Laboratory and also bred in-house. In all experiments inclusion criteria were healthy animals aged 6–12 weeks. There were no exclusions. Experiments were typically carried out 2–3 times with 5 mice per group to establish the reproducibility of the results and to accommodate processing and analysis of material. To reduce bias, mice in all experiments were matched for sex and age. For each experiment mice were allocated to their treatment group by randomization within blocks (Nuisance variables: sex, cage location). Groups were allocated by the person performing the experiment.

## Bacteria

*S. aureus* strain Newman and a streptomycin-resistant mutant of *S. aureus* strain Newman (Newman Sm$^R$) have been previously described [57]. Strains were grown from frozen stocks on tryptic soy agar (TSA) at 37˚C for 18h. All bacterial suspensions were prepared in sterile PBS and concentrations measured at an optical density of 600nm. CFUs were verified by plating serial dilutions of each inoculum onto TSA.

Human colonising strains RD9, RD21, RD25, RD44, from a cohort of persistent nasal carriers were provided by Prof Willem van Wamel. These isolates were collected as part of a previous study using an established protocol [58]. In brief, subjects were classified as persistent carriers when 3 consecutive nasal swab cultures (2 week intervals) were positive for *S. aureus*. In order to generate streptomycin resistant (Sm$^R$) mutants, these *S. aureus* human isolates were grown in TSB overnight, plated on Sm-supplemented plates and incubated overnight at 37˚C. Spontaneously generated Sm$^R$ colonies were then isolated and grown overnight on Sm-supplemented agar plates to confirm resistance.

## *S. aureus* nasal colonisation model

Mice received sterile water containing 0.5mg/ml streptomycin 48h before bacterial administration and for the duration of the experiment. Mice were inoculated intranasally (10μl/nostril) with *S. aureus* Newman Sm$^R$ (2x10$^8$CFU), human colonising strains where indicated (2x10$^8$CFU), or PBS alone. At specific time points post colonisation mice were culled. The area surrounding the nose was wiped with sterile ethanol wipes and the nose was excised by cutting approximately 1 cm deep from the outermost tip of the nose. The nasopharyngeal tissue (NT) which includes the nasopharynx-associated lymphoid tissue (NALT), consists of the upper palate and the paired NALT structures located bilaterally on the dorsal side of the floor of the nasal cavity, on the posterior side of the palate of the mouth in the mouse. To access the NT the lower jaw was removed, interior of the mouth cleaned with sterile ethanol wipes. The soft tissue was then separated from the rest of the nasal cavity by peeling away the ridged palate and removing the soft tissue at the dorsal side of the floor of the nasal cavity.

For IL-27 blocking studies; mice were administered with anti-IL-27 (150μg) (BioXcell) or isotype control IgG1 (150μg/nose) (BioXcell) at 24h prior to colonisation and again at the same time as bacterial inoculation.

For CFU enumeration noses and NT were homogenized in 500μl PBS and plated onto TSA with 0.5mg/ml streptomycin to quantify *S. aureus* colony forming unit (CFU)/ml. For protein measurements by ELISA noses and NT were homogenized in 500μl PBS and supernatants of

homogenates were used for analysis. For RT-PCR and flow cytometry analysis noses and NT were digested to isolate individual cells from the tissue.

Our studies require 10–20 animals/group to obtain statistically significant differences using two-tailed unpaired t-test, assuming $p<0.05$, 90% power, α error of 0.05. Power calculations using G-power software. Calculations based on previously published studies using models of *S. aureus* colonisation [14].

## Isolation of cells from the nasopharyngeal tissue

For flow cytometry and PCR analysis cells of the nose and NT were isolated using a digestion cocktail consisting of Collagenase XI (1mg/ml Sigma-Aldrich) Hyaluronidase (0.5mg/ml Sigma-Aldrich) DNase I (0.1mg/ml Sigma-Aldrich) for 30min at 37˚C in a shaking incubator, and filtered through a 40-μM nylon Falcon cell strainer. Cells were rested in "complete" RPMI media supplemented with L-glutamine (Sigma-Aldrich), penicillin-streptomycin (Sigma-Aldrich) and FBS (Sigma-Aldrich) for 30 mins in a 37˚C, 5% $CO_2$ incubator prior to mRNA isolation or analysis by flow cytometry.

## ELISA

ELISAs for IL-10 and IL-27 (R&D DuoSet; R&D Systems) were performed on cell culture supernatants, or supernatants of nose and nasal tissue homogenates as per the manufacturer's instruction.

## Flow cytometry

Isolated NT cells were incubated with PMA (50ng/ml) Ionomycin (500ng/ml) and Brefeldin A (5μg/ml) for 4h at 37˚C. Cells were then washed in PBS and stained with FixViability Dye eFluor506, followed by incubation in Fc block (αCD16/CD32) (ThermoFisher) before extracellular surface staining with fluorochrome-conjugated antibodies (Invitrogen unless stated) against CD45 (clone: 30-F11), CD3 (clone 145-2C11), CD4 (clone GK1.5), CD8 (clone 53–6.7), and γδ TCR (clone GL3). Cells were fixed and permeabilized, followed by intracellular staining with fluorochrome-conjugated antibodies against IL-17 (Biolegend; clone TC11-18H10.1) and IL-22 (clone 1H8PWSR). Fluorescence minus one (FMO) samples were used as controls. Flow cytometric data was acquired with a BD LSR Fortessa or BD FACSCanto II using Diva software (BD Biosciences) and analysed using FlowJo software (Tree Star, Inc).

## CD45+ MACS sorting and cell sorting

Cells of the nose and NT were further processed to isolate CD45+ cells by magnetic separation using a CD45 MACS kit (Miltenyi Biotec). The efficacy of the MACs sort was 95% or higher. For cell sorting experiments purified CD45+ cells were rested for 30 mins in a 37˚C, 5% CO2 incubator before surface staining with antibodies specific to CD3 (Invitrogen; clone 145-2C11), CD19 (Invitrogen; clone 1D3), CD11b (Invitrogen; clone M1/70), F480 (Invitrogen; clone BM8), Ly6G (Invitrogen; clone 1A8) and sorted into CD3[+], CD19[+], CD11b[+] F480[+], CD11b[+] Ly6G[+] cell populations using a BD FACsAria Fusion cell sorter. Cells were washed in sterile PBS and RNA was extracted from cells as described.

## RNA extraction, complementary DNA synthesis and qPCR

RNA from nose, NT, isolated nasal tissue leukocyte populations and BMDMs were extracted using Norgen total RNA purification microkit according to the manufacturer's instructions. RNA yields were measured on a BMG LABTECH SPECTRO star *Nano* machine. RNA

**Table 1. Primers used in qPCR.**

| Gene | Primer Pair | Supplier |
|---|---|---|
| *il10* | F: TTACCTGGTAGAAGTGATGC R: TAAAATCACTCTTCACCTGC | KiCqStart primers, Sigma-Aldrich |
| *Il27 p28* | F: ATCTCGATTGCCAGGAGTGAACCT R: AAATCCCAGCTCCCTCTCCTTTGT | Integrated DNA Technologies |
| *18s rRNA* | F: CCT GCG GCT TAA TTT GAC TC R: AAC TAA GAA CGG CCA TGC AC | Integrated DNA Technologies |

(250ng) was reverse transcribed using High-Capacity cDNA reverse transcription kit (Biosciences) according to the manufacturer's instructions. mRNA was quantified using quantitative PCR on a CFX96 Touch Real-Time pCR Detection System (Bio-Rad) using iTaq Sybr Green Supermix (Bio-Rad) according to the manufacturer's recommendations. Primer pairs are listed in Table 1. Expression of mRNA was calculated using the change-in-cycle threshold ($\Delta$ $\Delta$CT) method.

### *In vitro* culture of bone marrow derived macrophages (BMDMs) and infection with *S. aureus*

Murine BMDMs were cultured from WT or TLR2$^{-/-}$ mice using previously described protocols [59, 60]. On day 7 cells were seeded in a 96-well flat bottom plate and rested for 3h prior to assays. Cells were exposed to *S. aureus* Newman or human colonising strains at multiplicity of infection (MOI) 1,10 or 100 for 3h, then incubated in 100mg/ml gentamicin-supplemented DMEM for 1h to remove any extracellular bacteria, gentamicin-supplemented DMEM was removed and complete DMEM was added. Cells were then incubated for specified times of 4h or 24h. Supernatants were collected for analysis by ELISA, and cells were washed in PBS and RNA extracted as described.

In specific experiments cells were pre-treated with increasing concentrations of anti-IL-27 (BioXcell) or an appropriate isotype control (IgG2a, BioXcell) or recombinant IL-27 (BioLegend) 30 minutes prior to bacterial exposure. All media thereafter was supplemented with antibodies or recombinant IL-27 at the specified concentration.

### Statistical analysis

Statistical analyses were performed using GraphPad Prism 8 software (GraphPad Software, La Jolla, CA). For murine studies, differences between groups were analysed using an unpaired Student t test, one-way ANOVA with a Tukey comparison post-test, or two-way ANOVA with a Bonferroni correction post-test where appropriate. A p value $\leq 0.05$ was considered significant.

### Supporting information

**S1 Fig. B1a cells are present in the NT at 6h post-colonisation.** Wild-type mice were intranasally colonised with *S. aureus* Newman Sm$^R$ ($2 \times 10^8$ colony-forming units/nose). At 6h mice were culled, NT tissue was excised and digested for flow cytometry analysis. Cells were gated on single, live CD45+ cells> CD11b+> CD9+> CD5+. A representative FACs plot of this population is shown (Experimental unit = 1 mouse, total # mice used 5, experiment was performed once), indicating % cell populations.
(TIF)

**S2 Fig. BMDMs produce both IL-10 and IL-27 in response to *S. aureus* exposure.** WT BMDMs were exposed to *S. aureus* Newman at MOI 1, 10 or 100 for 3 h. Cells were washed with gentamicin to remove extracellular bacteria. Cells were then further incubated for 24 h

and levels of IL-10 (A) and IL-27 (B) were determined by ELISA. Values are expressed as mean protein concentration ± S.E.M. (Experimental unit = BMDMs isolated from 1 mouse, n = 8–10 per group, total # animals used 10, data generated from 8–10 independent experiments). Statistical analysis was carried out by one-way ANOVA, and student t-test. $^{**}P \leq 0.01$.
(TIF)

**S3 Fig. Human colonising strains induce IL-10 production and are capable of colonising mice.** WT BMDMs were exposed to *S. aureus* strains RD9, RD21, RD25, RD44 at MOI 10 for 3 h. Cells were washed with gentamicin to remove extracellular bacteria. Cells were then further incubated for 24 h and levels of IL-10 were determined by ELISA (A). Values are expressed as mean protein concentration ± S.E.M. (Experimental unit = BMDMs isolated from 1 mouse, n = 4 per group, total # animals used 4, data generated from 4 independent experiment). WT mice were intranasally colonised with *S. aureus* Newman Sm$^R$, RD21 Sm$^R$ and RD44 Sm$^R$ ($2 \times 10^8$ colony-forming units/nose). At 10 days mice were culled, noses were homogenized and serial dilutions of homogenates were plated onto streptomycin-supplemented TSA plates. Plates were grown overnight and CFUs were enumerated. Results are expressed as Log CFU/nose (B) and colonisation rate (C) as determined by the number of mice colonised by *S. aureus*/total number of mice. (Experimental unit = 1 mouse n = 3–6 mice per group, total # animals used 12, data generated from 2 independent experiment).
(TIF)

**S4 Fig. Bacterial burden is highest in the nose of mice post colonisation.** Wild-type mice were intranasally colonised with *S. aureus* Newman Sm$^R$ ($2 \times 10^8$ colony-forming units/nose). At 3 days (A) and 7 days (B) mice were culled and noses, nasopharyngeal tissue and lungs were excised. Tissue was homogenized and serial dilutions of homogenates were plated onto streptomycin-supplemented TSA plates. Plates were grown overnight and CFUs were enumerated. Results are expressed as CFU/organ (Experimental unit = 1 mouse n = 15 per group, total # animals used 30, data generated from 3 separate experiments,). Statistical analysis was carried out by one-way ANOVA, and student t-test. $^{**}P \leq 0.01$, $^{***}$ P$\leq$0.001.
(TIF)

**S5 Fig. The absence of IL-10 during nasal colonisation increases polymorphonuclear cell recruitment.** Wild-type and IL-10$^{-/-}$ mice were intranasally colonised with *S. aureus* Newman Sm$^R$ ($2 \times 10^8$ colony-forming units/nose). At 24h and 72h mice were culled, NT tissue was excised and digested for flow cytometry analysis. Cells were gated on single, live CD45+ cells> CD11b+> Ly6G+ cells. Results are expressed in absolute cell numbers (Ab #) with mean ± S.E.M. (Experimental unit = 1 mouse, n = 4 per group, total # animals used 24, data generated from 2 independent experiments). Statistical analysis was carried out by two-way ANOVA. $^{**}$P$\leq$0.01, $^{*}$P$\leq$0.05.
(TIF)

**S6 Fig. Human colonising strains induced IL-10 production is TLR 2 dependent.** WT BMDMs or TLR2-/- BMDMs were exposure to *S. aureus* strains RD25, RD44 at MOI 10 for 3 h. Cells were washed with gentamicin to remove extracellular bacteria and were then further incubated for 24 h and the levels of IL-10 were determined by ELISA. Values are expressed as mean protein concentration ± S.E.M. (Experimental unit = BMDMS isolated from 1 mouse, n = 3 per group, total # animals used 6, data generated from 3 independent experiments. Statistical analysis was performed using one-way ANOVA. $^{**}$ P$\leq$0.01, $^{***}$ P$\leq$0.001.
(TIF)

## Acknowledgments

We thank Willem van Wamel from Erasmus Medical Centre, Rotterdam for providing the human colonising strains.

## Author Contributions

**Conceptualization:** Alanna M. Kelly, John M. Leech, Rachel M. McLoughlin.

**Formal analysis:** Alanna M. Kelly, Rachel M. McLoughlin.

**Funding acquisition:** Rachel M. McLoughlin.

**Methodology:** Alanna M. Kelly, John M. Leech, Sarah L. Doyle.

**Project administration:** Rachel M. McLoughlin.

**Supervision:** Rachel M. McLoughlin.

**Writing – original draft:** Alanna M. Kelly.

**Writing – review & editing:** Sarah L. Doyle, Rachel M. McLoughlin.

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
