## [Decision Letter · Decision Letter 0]

28 Mar 2022

Dear Rachel,

Thank you very much for submitting your manuscript "Staphylococcus aureus induced immunosuppression mediated by IL-10 and IL-27- facilitates nasal colonization." for consideration at PLOS Pathogens. As with all papers reviewed by the journal, your manuscript was reviewed by members of the editorial board and by several independent reviewers. In light of the reviews (below this email), we would like to invite the resubmission of a significantly-revised version that takes into account the reviewers' comments.

All three reviewers requested the use of primary nasal or clinical S. aureus isolates in addition to the lab strain Newman and they asked for some additional control experiment. Moreover, they suggested modifications concerning the presentation and discussion of your data.

In addition to the reviewers’ comments we ask you to discuss potential mechanisms leading to IL-10 and IL-27-dependent loss of S. aureus from nares more thoroughly. You currently suggest that neutrophils, AMPs, and altered expression of epithelial attachment sites could impair S. aureus nasal colonization. However, there are probably not enough neutrophils in the nasal cavity to eradicate S. aureus directly and the nasal IgA cannot facilitate pathogen detection and phagocytosis. Moreover, AMPs are more or less equally active towards S. aureus and commensals and can hardly explain a selective inhibition of S. aureus. In the and, neutrophil and T-cell mediated mucosal inflammation may alter the abundance attachment sites and of nutrients on the nasal epithelium in a way that is unfavorable for sustained S. aureus colonization. We have problems though to think of a direct elimination process. Terms such as ‘clearance’ (at several positions in the manuscript) or ‘eradication’ (lines 84, 379) may not be appropriate as long as there is no evidence for direct elimination.

We cannot make any decision about publication until we have seen the revised manuscript and your response to the reviewers' comments. Your revised manuscript is also likely to be sent to reviewers for further evaluation.

Sincerely,

Andreas Peschel, Ph.D.

Associate Editor

PLOS Pathogens

Michael Otto

Section Editor

PLOS Pathogens

Kasturi Haldar

Editor-in-Chief

PLOS Pathogens

orcid.org/0000-0001-5065-158X

Michael Malim

Editor-in-Chief

PLOS Pathogens

orcid.org/0000-0002-7699-2064

Reviewer's Responses to Questions

**Part I - Summary**

Reviewer #1: In this study, Kelly et al. sought to explore the immunological basis behind Staphylococcus aureus persistent nasal colonization. The authors focus their attention on production of IL-10 and IL-27 in the nose and nasopharyngeal tissue, which leads to reduced inflammatory T cell responses and consequently reduces the efficiency of clearance. Using a murine colonization model and in vitro macrophage assays, the authors determine that S. aureus induces the production of IL-10 from CD45+ cells (primarily B cells, macrophages, and MDSC). The production of IL-10 correlates with increased colonization persistence as S. aureus is cleared more rapidly in IL-10-/- mice. The rapid clearance was correlated with heightened production of IL-17 and IL-22, cytokines previously implicated in improved efficiency of bacterial clearance. The authors further demonstrate that IL-27 production leads to increased IL-10 production, potentially via autocrine signaling. Antibody-mediated inhibition of IL-27 reduces the levels of IL-10 and consequently enhances bacterial clearance in the nares. Lastly, the TLR2 receptor was implicated as the primary mediator of IL-10 production that is enhanced via IL-27. Overall, the manuscript addresses a relevant question in the S. aureus field, providing a significant advance in our understanding of the interplay between the bacterium and the host immune system. The experiments are for the most part appropriately designed, and include relevant controls . In most instances, the conclusions are supported by the data. There are a few concerns related to strain choice and data interpretation that should be addressed by the authors.

Reviewer #2: Overall this is a well-constructed study with data supporting a role for S. aureus-mediated induction of IL-10 via IL-27 in slowing the rate of S. aureus nasal clearance in a mouse model of nasal colonization. The experiment blocking IL-27 in vivo in WT vs IL-10-/- mice with intranasal administration of anti-IL-27 antibody compared with isotype provides particularly strong support that IL-27 induces the IL-10 response to S. aureus Newman in the nose. (It also suggests that perhaps the key events are occurring in the mucus layer/lumen.)

Reviewer #3: Kelly et al present a manuscript investigating the role of the anti-inflammatory cytokine IL-10 during nasal colonisation of S. aureus.

The authors find that IL-10 levels in the nasopharyngeal tissue of mice increase upon colonisation with S. aureus and identified B-cells, and myeloid cells as a source of IL-10. Increased levels of IL-10 entailed, that the numbers of leukocytes creating an inflammatory environment (by the secretion of IL-17 and IL-22) decreased. Additionally the authors show that also IL-27 production is important for stimulating IL-10 production. IL-27 was identified to act upstream of Il-10 in an autocrine fashion. Finally, in agreement with earlier publications the authors show that IL-10 production upon S. aureus colonisation depends on stimulation of TLR-2 dependent signalling.

This reviewer is a molecular microbiologist accordingly detailed assessment of the experiments in IL-dependent inflammatory cascades have to be referred form other reviewers. However, in my opinion this manuscripts draws a comprehensive picture of the inflammatory surrounding created within the nasal cavity upon colonisation of S. aureus. The host-immunity associated with nasal colonisation is in general neglected and this study represents in important extension to our current understanding.

**Part II – Major Issues: Key Experiments Required for Acceptance**

Reviewer #1: 1. The authors conduct all studies with strain Newman. In principle this is fine, however it would be informative to know if the observed phenotypes occur in a non-Newman strain. At minimum, the authors should discuss this limitation to the study.

2. The evidence in support of upregulation of il10 transcript in B cells, macrophages, and MDSC (Fig 1F) is reasonably compelling (although no stats were provided – see below). However, the same data in Figure 4F for il27 is not so strong and hovers right at the cut-off level. The authors comment confidently that the sources of IL-27 are these cell types (and is part of the basis for their autocrine regulation hypothesis), but it is not clear the data fully support this conclusion. Could there be another CD45+ cell type in vivo that is producing IL27? The data from the figure with purified BMDM supports the autocrine model, but this may not be what happens in vivo? The authors may consider adjusting their interpretations or discussing this alternative possibility in light of the strength of the data.

3. Lines 228 to 230. The simplest and most accurate interpretation of the data is that the IL-27 signal peaks prior to IL-10. One possibility, of many, is that IL-27 could act upstream to impact levels of IL-10. The authors will test this possibility in the following section, but they should consider only stating what can be firmly concluded from the results in Figures 1 and 4 until they have data to support this specific possibility.

4. It is suprising that no time points earlier than three days were examined in the nasal colonization comparisons between WT and IL10-/- mice. Experiments in figure 1 look as early as six hours and begin to show differences in IL-10 and IL-27. What does colonization look like at very early timepoints, maybe 6 or 12 hours? This information would also be helpful to establish whether or not S. aureus ever initially colonizes an IL10-/- mouse (by day 3 there are very few bacteria remaining).

Reviewer #2: The focus entirely on a single, domesticated strain of S. aureus is a downside to a well-constructed set of experiments that support a role of S. aureus-mediated induction of IL-10 in preventing more rapid nasal clearance in a mouse nasal colonization model. Use of a primary nostril isolate, or clinical isolate, rather than Newman, or, at the least, validation of a key finding with a primary nasal isolate in at least one experiment would strengthen the manuscript and broaden the implications of the findings.

Reviewer #3: However, I have two mayor concerns regarding the experimental procedures. It is well appreciated that S. aureus strains possess strikingly different abilities to cause inflammation, this depends on the activity of various cellular machineries. The authors conducted their experiments using the single strain S. aureus Newman and verified in their experiments, that TLR 2-signalling is important for IL-10 production. TLR2 stimulation differs between lineages. For example lipoprotein shedding is shaped by agr-activity and their degradation by lipases can dampen TLR2 activation (Chen and Alonzo 2019). In strain Newman a phospholipase is inactivated by phage integration (Bea…Schneewind, 2006), entailing increased TLR2 activation. The authors should verify (at least in vitro) using a panel of S. aureus strains that the effects of S. aureus on IL-10 production are widely conserved between strains.

Secondly, the authors should attempt to verify their experiments in humans. It might be possible to determine IL-10 levels in the nasal cavities of humans colonized with S. aureus vs humans lacking S. aureus.

**Part III – Minor Issues: Editorial and Data Presentation Modifications**

Reviewer #1: 1. Line 61: Consider changing microflora to microbiota.

2. Lines 171-172: The authors should stick with one convention for colonised/colonized. It is used interchangeably in places.

3. Line 321. UMAS-1 should be UAMS-1.

4. Were statistics done on Figs 1E-F and Fig 4E-F? Some reasonably strong conclusions are made about these data. If statistical significance was not achieved, the authors should appropriately consider toning things down and pointing out that statistical significance was not achieved.

5. Please define the abbreviation Ab# in the figure legend of figure 3.

Reviewer #2: A major question that could be at least raised in the Discussion is whether this host immunological response is specific to S. aureus nasal colonization or does it occur with other more common human-associated nasal bacteria, e.g., Staphylococcus epidermidis or nasal-associated non-diphtheriae Corynebacterium species. Similarly, do other common nasal colonizers, many of which have higher prevalence in the human population than S. aureus, alter the host IL-27 and IL-10 response? In the long run it seems key to determine whether the possibility of altering the host response by dampening IL-10 to increase S. aureus clearance would also impact more benign nasal colonizers in an undesirable manner

MODERATE ISSUES

For Fig. 1, why were nose and nasopharyngeal tissue analyzed separately? Please add this information to the Results section. Also, please indicate in the Methods, lines 441-444, how, and using what anatomic landmarks, the nose and nasopharyngeal tissue were separated from each other for analysis.

Since S. aureus colonization burden is highest in the nose in this mouse model (Fig S1), it is unclear why the analysis for IL-22+ and IL-17+ T cells (Fig. 3) was done using nasopharyngeal tissue rather than nose tissue.

In comparing Figure 4 (IL-27) and Figure 1 (IL-10), for which the same animals were used, what might be inducing an increase in IL-10 in the nose (Fig 1D) in S. aureus colonized animals in the absence of a detectable induction of IL-27 (Fig 4D)?

Abstract & Line 62: 30% persistent S. aureus colonization picks the high side of what exists in different studies and is very likely an overestimate. A range or a mid-point would be more appropriate.

Is the mechanism responsible for the almost 5 log drop in S. aureus from the 2x10^8 inoculum to ~4x20^3 level on Day 3 (Fig. 2) known?

MINOR ISSUES

Please clarify in the abstract that data are based on the use of a mouse model of S. aureus nasal colonization.

Please clarify the # of animals needed per group which is listed in Line 423 as 5 animals/group/experiment x2-3 experiments, which is 10 or 15 per group, but in Line 456 as ~15 animals/group.

Please include an indication of how many independent experiments were done for panels A-D in Figures 1 & 4 as well as for Figure 2, Figure 3, Figure 5, and Fig S2.

Any thoughts as to why the IL-10 protein levels at 24 hours in Figure 5 are greater than 4 times lower than those in Figure 1?

Is there a typo on line 804 regarding the total animals used?

Consider using microbiota rather than microflora, as the former is technically more accurate.

Line 165: define the abbreviation MDSC when first used.

Reviewer #3: Fig. E the colour of the bars and the legend seems not to match.

Fig4 F: The expression levels in the B-cells and myeloid cells appears not to differ significantly from the baseline. Does this result really allow the interpretation that these cells produce the IL upon S. aureus colonisation?

The result shown in FigS3 is describe discussion the first time. It should be moved to the results part.-

PLOS authors have the option to publish the peer review history of their article (what does this mean?). If published, this will include your full peer review and any attached files.

Reviewer #1: No

Reviewer #2: No

Reviewer #3: No
---

## [Decision Letter · Decision Letter 1]

6 Jun 2022

Dear Ms Kelly,

We are pleased to inform you that your manuscript 'Staphylococcus aureus induced immunosuppression mediated by IL-10 and IL-27 facilitates nasal colonisation.' has been provisionally accepted for publication in PLOS Pathogens.

Best regards,

Andreas Peschel, Ph.D.

Associate Editor

PLOS Pathogens

Michael Otto

Section Editor

PLOS Pathogens

Kasturi Haldar

Editor-in-Chief

PLOS Pathogens

orcid.org/0000-0001-5065-158X

Michael Malim

Editor-in-Chief

PLOS Pathogens

orcid.org/0000-0002-7699-2064

Reviewer Comments (if any, and for reference):

Reviewer's Responses to Questions

**Part I - Summary**

Reviewer #1: The authors have addressed previous concerns by including new data and revised text. They have included an assessment of several colonizing strains for IL-10 production by BMDM in vitro and two for colonization levels in vivo. There appear to be comparable responses and colonization levels. The authors also included a revised figure 2 that helps to strengthen some conclusions. The edits to the text and conclusions were felt to be appropriate and improved the overall manuscript.

Reviewer #2: From my perspective, the authors have done a good job in addressing reviewer comments and concerns in this revised manuscript.

Reviewer #3: (No Response)

**Part II – Major Issues: Key Experiments Required for Acceptance**

Reviewer #1: None.

Reviewer #2: (No Response)

Reviewer #3: (No Response)

**Part III – Minor Issues: Editorial and Data Presentation Modifications**

Reviewer #1: None.

Reviewer #2: (No Response)

Reviewer #3: (No Response)

PLOS authors have the option to publish the peer review history of their article (what does this mean?). If published, this will include your full peer review and any attached files.

Reviewer #1: No

Reviewer #2: No

Reviewer #3: No

---

## [Editor Report · Acceptance letter]

28 Jun 2022

Dear Dr McLoughlin,

We are delighted to inform you that your manuscript, "*Staphylococcus aureus*-induced immunosuppression mediated by IL-10 and IL-27 facilitates nasal colonisation.," has been formally accepted for publication in PLOS Pathogens.

Best regards,

Kasturi Haldar

Editor-in-Chief

PLOS Pathogens

orcid.org/0000-0001-5065-158X

Michael Malim

Editor-in-Chief

PLOS Pathogens

orcid.org/0000-0002-7699-2064